# Isoflavone Aglycones Attenuate Cigarette Smoke-Induced Emphysema via Suppression of Neutrophilic Inflammation in a COPD Murine Model

**DOI:** 10.3390/nu11092023

**Published:** 2019-08-29

**Authors:** Kazuya Kojima, Kazuhisa Asai, Hiroaki Kubo, Arata Sugitani, Yohkoh Kyomoto, Atsuko Okamoto, Kazuhiro Yamada, Naoki Ijiri, Tetsuya Watanabe, Kazuto Hirata, Tomoya Kawaguchi

**Affiliations:** Department of Respiratory Medicine, Graduate School of Medicine, Osaka City University, Osaka 545-8585, Japan

**Keywords:** daidzein-rich soy isoflavone aglycones (DRIAs), COPD, neutrophilic inflammation, TNF-α, C-X-C motif ligand 2 (CXCL2)

## Abstract

Chronic obstructive pulmonary disease (COPD), a lung disease caused by chronic exposure to cigarette smoke, increases the number of inflammatory cells such as macrophages and neutrophils and emphysema. Isoflavone is a polyphenolic compound that exists in soybeans. Daidzein and genistein, two types of isoflavones, have been reported to have anti-inflammatory effects in various organs. We hypothesized that the daidzein-rich soy isoflavone aglycones (DRIAs) attenuate cigarette smoke-induced emphysema in mice. Mice were divided into four groups: the (i) control group, (ii) isoflavone group, (iii) smoking group, and (iv) isoflavone + smoking group. The number of inflammatory cells in the bronchoalveolar lavage fluid (BALF) and the airspace enlargement using the mean linear intercept (MLI) were determined 12 weeks after smoking exposure. Expressions of neutrophilic inflammatory cytokines and chemokines were also examined. In the isoflavone + smoking group, the number of neutrophils in BALF and MLI was significantly less than that in the smoking group. Furthermore, the gene-expressions of TNF-α and CXCL2 (MIP-2) in the isoflavone + smoking group were significantly less than those in the smoking group. Supplementation of the COPD murine model with DRIAs significantly attenuates pathological changes of COPD via suppression of neutrophilic inflammation.

## 1. Introduction

Chronic obstructive pulmonary disease (COPD) causes chronic obstruction of lung airflow, and is characterized by emphysematous changes and peripheral airway lesions. Many patients suffer from dyspnea, cough, and shortness of breath. COPD is the third leading cause of death in the world [1]. Cigarette smoke is the most important risk factor for the development of COPD. Neutrophilic inflammation is one of the characteristics of COPD, and airway neutrophilia is associated with a decline in lung function and leads to pulmonary emphysema [2,3]. Pulmonary emphysema is progressive, and hence the treatment or prevention of emphysema is desired. Chronic exposure to cigarette smoke increases cytokine secretion and expression of pro-inflammatory genes such as tumor necrosis factor-α (TNF-α), granulocyte-colony stimulating factor (G-CSF), C-X-C motif ligand 1 (CXCL1; KC), or C-X-C motif ligand 2 (CXCL2; MIP-2), resulting in neutrophil inflammation. In particular, CXCL1 (KC) and CXCL2 (MIP-2) play an important role in neutrophil recruitment to sites of inflammation and tissue injury [4,5,6,7,8]. 

Reactive oxygen species (ROS) cause oxidative stress, and promote the recruitment of neutrophils and other inflammatory cells. ROS also induces the release of pro-inflammatory mediators that promote inflammation, which can contribute to the development of emphysema. Oxidative stress, in particular caused by neutrophilic inflammation, is an important factor in COPD pathogenesis [9]. We previously reported that the expression of nuclear factor erythroid 2-related factor 2 (Nrf2), a regulator of antioxidant defense, was lower in human bronchial epithelial cells in COPD [10]. Another report showed Nrf2 activation reduced the oxidative stress caused by cigarette smoke in the lung tissues [11]. Therefore, COPD patients are susceptible to oxidative stress induced epithelial cell apoptosis leading to emphysema. 

Isoflavone is a polyphenolic compound that exists in a number of foods, including soybeans, and daidzein and genistein are the major types of isoflavones. Daidzein is reported to possess anti-inflammatory effects in various organs [12,13,14,15,16]. A recent report shows that isoflavone plays an important role as a scavenger of ROS generated by human neutrophils [14]. The consumption of total soy is reported to be positively correlated with lung function measures in COPD [17]. Additionally, the risk of COPD is lower in people who consume high doses of soybean products daily [17,18]. These reports suggested that soy isoflavones have an anti-inflammatory effect and protect against cigarette smoke-induced inflammation. However, the underlying mechanism of the protective effects of isoflavones in COPD is still unknown.

In this study, we hypothesized that daidzein-rich soy isoflavone aglycones (DRIAs) attenuate pulmonary emphysema caused by chronic exposure to cigarette smoke in the COPD murine model. We also examined results from the gene-expression analysis related to neutrophilic inflammation.

## 2. Materials and Methods

### 2.1. Animals and Supplementation with DRIAs

Four-week-old male C57BL/6 mice were purchased from Japan SLC (Shizuoka, Japan) and were maintained at a temperature of 23 ± 2 °C under 12 h/12 h day/night cycles. Mice were randomly divided into four groups: (i) a control group (*n* = 7) that consisted of non-smoking mice on a normal diet (MF diet), (ii) an isoflavone group (*n* = 10) that consisted of non-smoking mice on an MF diet containing 0.6% DRIAs including daidzein, genistein, and glycitein (AglyMax; Nichimo Co. Ltd., Tokyo, Japan), (iii) a smoking group (*n* = 8) that consisted of smoking mice on an MF diet, and (iv) an isoflavone + smoking group (*n* = 8) that consisted of smoking mice on an MF diet containing 0.6% DRIAs. We used an MF diet that contained 7.9% of water, 23.1% of proteins, 5.1% of fat, 5.8% of minerals, 2.8% of fibers, and 55.3% of carbohydrates. The DRIAs contained 323 mg total isoflavone /g: 202 mg daidzein, 31 mg genistein, and 90 mg glycitein. These diets were prepared by Oriental Yeast Co. Ltd. (Tokyo, Japan). After randomization, mice started to receive either the MF diet or the MF diet with DRIAs. The timing of receiving each diet was prior to the cigarette smoke challenge. Animal experiments were approved by the Ethics Committee of the Institutional Animal Care and Use (approval number 17023). 

### 2.2. Chronic Exposure to Cigarette Smoke

After acclimatization for 1 week, the smoking groups were exposed to cigarette smoke using a cigarette smoke generator-Model SG-300 (Shibata Scientific Technology. Tokyo, Japan) for 60 min/day and 5 days/week for 12 weeks. Peace^®^ (Japan Tobacco, Inc., Tokyo, Japan), a commercially marketed, nonfilter cigarette containing 28 mg of tar and 2.3 mg of nicotine per cigarette was used. 

### 2.3. Bronchoalveolar Lavage (BAL)

After the 12-week intervention period, mice were sacrificed. Bronchoalveolar lavage (BAL) was performed three times on inflated lungs using 0.5 mL of phosphate-buffered saline. The bronchoalveolar lavage fluid (BALF) was centrifuged and cells were resuspended. The total cell and differential cell counts were determined in each specimen by counting more than 400 cells. The supernatant of the BALF was stored at −80 °C for further examination. 

### 2.4. Lung Histology

The left lungs were fixed using 10% formalin solution at a constant pressure of 25 cm H_2_O for 48 h. Samples were embedded in paraffin, sectioned at a thickness of 3 µm and stained with hematoxylin and eosin. The airspace enlargement was evaluated by measuring the mean linear intercept (MLI) as previously described [19]. 

### 2.5. Cytokine and Nrf2 Expression Analysis

The right lungs were soaked in Buffer RLT (Qiagen, Tokyo, Japan) and homogenized using a Bead Mill 24 (Thermo Fisher Scientific, Waltham, Tokyo, Japan). Homogenized samples were centrifuged at 12,000 rpm for 3 min, and total RNA was extracted using an RNeasy mini kit (Qiagen, Tokyo, Japan). RNA concentrations were evaluated using a NanoDrop (Thermo Fisher Scientific, Tokyo, Japan). Complementary DNA was prepared using a Ready-to-Go-T-primed first-strand kit (GE Healthcare, Little Chalfont, UK). Quantitative real-time PCR was performed using an Applied Biosystems 7500 Real-Time PCR System (Thermo Fisher Scientific, Tokyo, Japan). The PCR mixture contained 1 µL of cDNA samples and primers of each of the target genes. TaqMan gene expression assays (Thermo Fisher Scientific, Tokyo, Japan) for CXCL1 (Mm04207460_m1), CXCL2 (Mm00436450_m1), TNF-alpha (Mm00443258_m1), G-CSF (Mm00432735_m1), Nrf2 (Mm_00477784_m1), and 36B4 (Mm00725448_s1) were performed. We used 36B4 as an internal control gene and the relative levels of each mRNA were normalized to 36B4 mRNA levels using the ΔΔCT method.

### 2.6. Enzyme-Linked Immunosorbent Assay (ELISA)

The concentrations of CXCL1 (KC) and CXCL2 (MIP-2) in the supernatant of the BALF were measured using the commercially available ELISA kit, in accordance with the manufacturer’s instructions. The Mouse KC ELISA Kit (MyBiosource, San Diego, CA, USA) and the Mouse CXCL2 (MIP-2) Quantikine ELISA Kit (R&D Systems, Minneapolis, MN, USA) were used for the detection of CXCL1 (KC) and CXCL2 (MIP-2), respectively. The absorbance was measured at 450 nm.

### 2.7. Statistical Analysis

Data were expressed as the mean ± SD, unless otherwise stated. Data were normally distributed and analyzed using one-way ANOVA, followed by multiple comparisons using Bonferroni’s procedure. In all statistical analyses, *p* < 0.05 was considered significant.

## 3. Results

### 3.1. Changes in Body Weight

Body weight for each group measured weekly is shown in Figure 1. Body weights steadily increased during the 12 weeks. The increase in body weights of the two smoking groups (smoking and isoflavone + smoking group) was significantly less than the two non-smoking groups (control and isoflavone group). There was no significant difference between the mice in the MF containing DRIA group and the mice in the MF group in both smoking and non-smoking settings. 

### 3.2. Effect of DRIAs on the Pathological Changes of COPD 

Representative BALF microscopic images are shown in Figure 2A–D. The number of total cells in BALF in the two smoking groups (smoking and isoflavone + smoking group) was significantly higher than the two non-smoking groups (control and isoflavone group), while there were no significant differences in the number of total cells between the control group and isoflavone group, and between the smoking group and isoflavone + smoking group (Figure 2E). The number of macrophages in the BALF in the smoking group was significantly higher than the control group (Figure 2F). Moreover, the number of lymphocytes in the BALF in the smoking groups were also significantly higher than the non-smoking groups (Figure 2G). There were no significant differences in the number of macrophages and lymphocytes in the BALF between the control group and isoflavone group, and between the smoking and isoflavone + smoking group. However, the number of neutrophils in the BALF in the smoking groups was significantly higher than the non-smoking groups, and the number of neutrophils in the isoflavone + smoking group was significantly less than the smoking group (Figure 2H).

Representative hematoxylin and eosin staining microscopic images are shown in Figure 3. In the smoking group, marked airspace enlargement and the loss of alveolar walls were observed. The MLI in the smoking group was significantly higher than the control group. Moreover, the MLI in the isoflavone + smoking group was significantly less than the smoking group, and its value was as small as that in the control group (Figure 3E).

### 3.3. Effect of DRIAs on Inflammatory Mediators in the Lungs

We assessed the CXCL1 (KC), CXCL2 (MIP-2), TNF-α, G-CSF, and Nrf2 mRNA levels in the lung homogenate using quantitative real-time PCR. The gene expression of CXCL1 (KC) in the smoking groups was significantly higher than the control group (Figure 4A). However, there was no significant difference in the gene expression of CXCL1 (KC) between the smoking group and isoflavone + smoking group. On the other hand, the gene-expressions of CXCL2 (MIP-2) and TNF-α in the smoking groups were significantly higher than the control group (Figure 4B,C). In addition, the gene-expressions of CXCL2 (MIP-2) and TNF-α in the isoflavone + smoking group were significantly less than the smoking group. There were no significant differences between groups in the gene-expressions of G-CSF and Nrf2, as an antioxidative marker (Figure 4D,E).

### 3.4. ELISA for CXCL1 (KC) and CXCL2 (MIP-2) in BALF

There were no significant differences between groups in terms of CXCL1 (KC) expression in BALF (Figure 5A). However, the CXCL2 (MIP-2) expression in BALF in the smoking groups was significantly higher than the control group (Figure 5B). The CXCL2 (MIP-2) expression in the isoflavone + smoking group was significantly less than the smoking group.

## 4. Discussion

In this study, we showed that isoflavone supplementation in cigarette smoke-induced COPD in murine models significantly attenuated the neutrophilic inflammation via suppression of mRNA levels of TNF-α and CXCL2 (MIP-2). Isoflavone also almost completely attenuated the MLI increase. This result supports our hypothesis that treatment with DRIAs significantly suppresses cigarette smoke-induced pulmonary emphysema. This is the first report stating that soy isoflavone has a protective effect on emphysema formation in an in vivo model.

Isoflavones exist as either glucoside or aglycone forms, and isoflavone glucosides are generally known to be converted to the aglycone forms by gut microflora or gut glucosidases. The aglycone forms are absorbed more efficiently than the glucoside forms. Isoflavone aglycones are found primarily in soy products such as miso, natto, and soy sauce [20]. Epidemiologically, the consumption of soy is reported to be positively correlated with lung function measures and soybean products have protective effects on COPD onset [17,18], while the details of the mechanism are not clearly understood. In general, both emphysema and peripheral airway lesions are considered to work cooperatively in COPD pathogenesis, and our results present a proposed mechanism for the epidemiological observation of the relation between soy isoflavone and COPD.

Neutrophils are well recognized as leukocytes that are recruited to sites of inflammation to protect against pathogens such as bacteria and fungus [21]. Neutrophils are activated by bacterial products, cytokines, or chemokines, e.g., TNF-α, INF-γ, CXCL1 (KC) and CXCL2 (MIP-2). They destroy pathogens by multiple mechanisms that involve proteases such as neutrophil elastase (NE). Exposure to cigarette smoke is a trigger for the release of multiple substances like TNF-α from epithelial cells and alveolar macrophages that recruit inflammatory cells to the lungs [22,23,24]. NE released from activated neutrophils by cigarette smoke-induced inflammation is important for killing bacteria and fungus, but also causes damages to lung tissues. NE can degrade connective tissue proteins such as elastin, collagen, and proteoglycan [25,26,27]. NE also causes direct epithelial damage [28]. In response to cigarette smoke, inflammatory cells such as neutrophils and macrophages are recruited to the lungs and release cytokines and proteolytic enzymes, causing the destruction of the lung tissues. These changes related to neutrophilic inflammation are involved in the pathogenesis of COPD. Intervening in neutrophil recruitment is a key strategy against emphysema formation.

In previous studies, soy isoflavones have been suggested to have an anti-inflammatory effect [12,13,14,15,16]. As previously described, it is known that neutrophilic inflammation is the main inflammatory feature in COPD airways [3,26]. In this study, we showed that isoflavone supplementation to cigarette smoke-induced COPD murine models significantly attenuates neutrophilic inflammation. To unveil the underlying mechanism, we examined expression of neutrophil chemoattractants in the lung. Chronic exposure to cigarette smoke increases cytokine secretion and pro-inflammatory gene expression. TNF-α is a general pro-inflammatory marker and CXCL1 (KC), CXCL2 (MIP-2), or G-CSF are also known to be potent chemoattractants for neutrophils. Macrophages synthesize and release CXCL1 (KC) and CXCL2 (MIP-2). Both chemokines are major ligands for C-X-C chemokine receptor 2 (CXCR2) but have different affinities for it. We examined mRNA expressions of these four markers in lung homogenates. Although CXCL1 (KC) is known to be the most potent chemokine for neutrophil chemotaxis, supplementation with DRIAs did not change the CXCL1 (KC) mRNA expressions. However, for another neutrophil chemokine, CXCL2 (MIP-2), supplementation with DRIAs caused significantly lower mRNA expression. We also examined the expression of these two neutrophil chemokines in BALF by ELISA. Although the protein levels of CXCL1 (KC) did not decrease, those of CXCL2 (MIP-2) significantly decreased as a result of isoflavone supplementation in BALF concordant with their mRNA expressions. This discrepancy in chemokine expression is similar to that found in previous reports on the effect of daidzein on inflammatory mediators [13]. This report showed that isoflavones attenuated CXCL2 (MIP-2) expression. Poly-adenosine diphosphate ribosylation (PARsylation) is a process of formation of poly-adenosine diphosphate ribose (PAR). PARP-1 is one of the PAR polymerases and has 85–90% of PAR polymerase activity [29]. Cigarette smoke induces DNA damage associated with PARP-1 activation [30]. The activation of PARP-1 is a trigger for enhancing the expression of pro-inflammatory cytokines. PARP-1 modulates NF-kB activity, thus leading to the activation of cytokines such as TNF-α and IL-1β [13,14,30]. TNF-α and IL-1β increase CXCL2 (MIP-2) expression. Daidzein is known to attenuate PARP-1 activity, and subsequently inhibit NF-kB and CXCL2 (MIP-2) expression [13]. Our results show that the suppression of CXCL2 (MIP-2) levels by DRIAs might be related to the attenuation of PARP-1 activity.

A previous study showed the differences between CXCL1 (KC) and CXCL2 (MIP-2) expressions in an in vivo model [31]. It reported that the number of neutrophils in the junctional epithelium of specific-pathogen-free (SPF) mice was higher than that in germ-free (GF) mice. Although there were no differences in CXCL1 (KC) expression levels between SPF and GF mice, CXCL2 (MIP-2) expression levels in SPF mice were different from that in GF mice. CXCL2 (MIP-2) expression was suggested to be regulated by oral commensal bacterial colonization [32]. Lipopolysaccharide (LPS) is a bacterial endotoxin that exists in the outer membrane of gram-negative bacteria, and cigarette smoke also contains LPS. LPS is an active component of cigarette smoke and has effects on lung tissue [33]. LPS inhalation induces CXCL2 (MIP-2) expression, and isoflavones are reported to reduce CXCL2 (MIP-2) expression [34]. Our results show that DRIAs suppressed CXCL2 (MIP-2) levels, but not CXCL1 (KC) levels, which might be related to their response to LPS in cigarette smoke. Detailed study regarding the differences between CXCL1 (KC) and CXCL2 (MIP-2) expression in response to cigarette smoke and DRIAs is needed.

Another previous study reported another anti-inflammatory mechanism of flavonoids. Phytoestrogens, which contain daidzein and genistein, are known to increase dehydroepiandrosterone sulfate (DHEA-S) synthesis [35], which in turn inhibits human neutrophil migration [36]. In our study, we measured DHEA-S concentration in the serum of mice using an enzyme immunoassay (EIA) kit. However, DHEA-S concentrations were undetectable, and we could not find a correlation with neutrophils in BALF. Our data showed that the pathway of DHEA-S synthesis was not associated with the decreased effect of DRIAs on neutrophils in this study.

Nrf2 is known as a transcription factor responsible for antioxidant capacity. The activation of the Nrf2 signaling pathway is a major mechanism in the cellular defense against oxidative stress [37]. A previous report showed that Nrf2-deficient mice were highly susceptible to cigarette smoke-induced emphysema [38,39]. We also previously reported that Nrf2 expression in airway epithelial cells of COPD patients was significantly less than that in healthy subjects [10], and COPD patients were also susceptible to cigarette smoke-induced emphysema. Flavonoids such as daidzein and genistein are known to be one of the inducers of Nrf2 [40], therefore, we examined the mRNA expression of Nrf2 in mice lungs as an antioxidant marker. However, there were no significant differences between the control and isoflavone group. We could not find a significant association between DRIAs and the mRNA level of total Nrf2 in this study.

Our study had some limitations. First, although we showed that DRIAs decrease the number of neutrophils in the BALF and simultaneously attenuate pulmonary emphysema, we did not evaluate the direct relationship between neutrophilic inflammation and emphysema. However, previous studies have reported a strong relationship between them. Secondly, COPD is more common in men, and male SPF mice were used in this study as an in vivo model. However, we did not discuss any gender-specific response. Isoflavone is absorbed from the gut and transformation in the gut is influenced by gut flora. However, the gut flora is different in humans and SPF mice. Thirdly, we did not measure plasma levels of daidzein and genistein in the mice before and after supplementation. Therefore, we could not examine the bioavailability of isoflavone and dose-response of isoflavone on inflammation. Fourthly, it is unclear whether the isoflavone dose used in this study is relevant for humans, while previous studies reported that 0.5% of soy isoflavone in the diet for mice is similar to a dose in humans of 810 mg isoflavone/50 kg body weight per day, and it is the amount that humans consume as a dietary supplement [41,42,43]. Further studies are warranted.

## 5. Conclusions

In conclusion, 0.6% DRIAs significantly attenuate cigarette smoke-induced emphysema by at least partially attenuating neutrophilic inflammation. The attenuation is probably via suppression of TNF-α and CXCL2 (MIP-2). This result extends a new insight into COPD pathogenesis and a way for a new strategy for COPD treatment and prevention.

## Figures and Tables

**Figure 1 nutrients-11-02023-f001:**
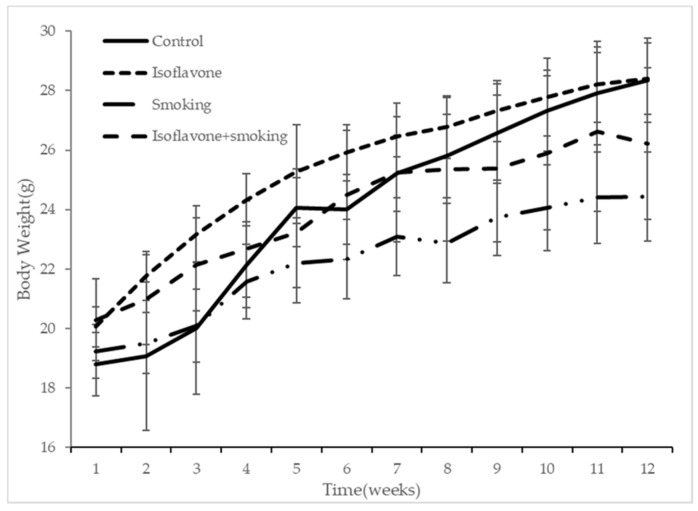
Body weight changes in each group. Control group; non-smoking with the normal diet (MF diet), (*n* = 7), isoflavone group; non-smoking with the MF diet with DRIAs (daidzein-rich soy isoflavone aglycones), (*n* = 10), smoking group; smoking with the MF diets, (*n* = 8), and isoflavone + smoking group; smoking with the MF diet with DRIAs, (*n* = 8). Values represent means ± SD. There are significant differences in the body weight between the control group and isoflavone group, and the smoking group and isoflavone group after 2 weeks.

**Figure 2 nutrients-11-02023-f002:**
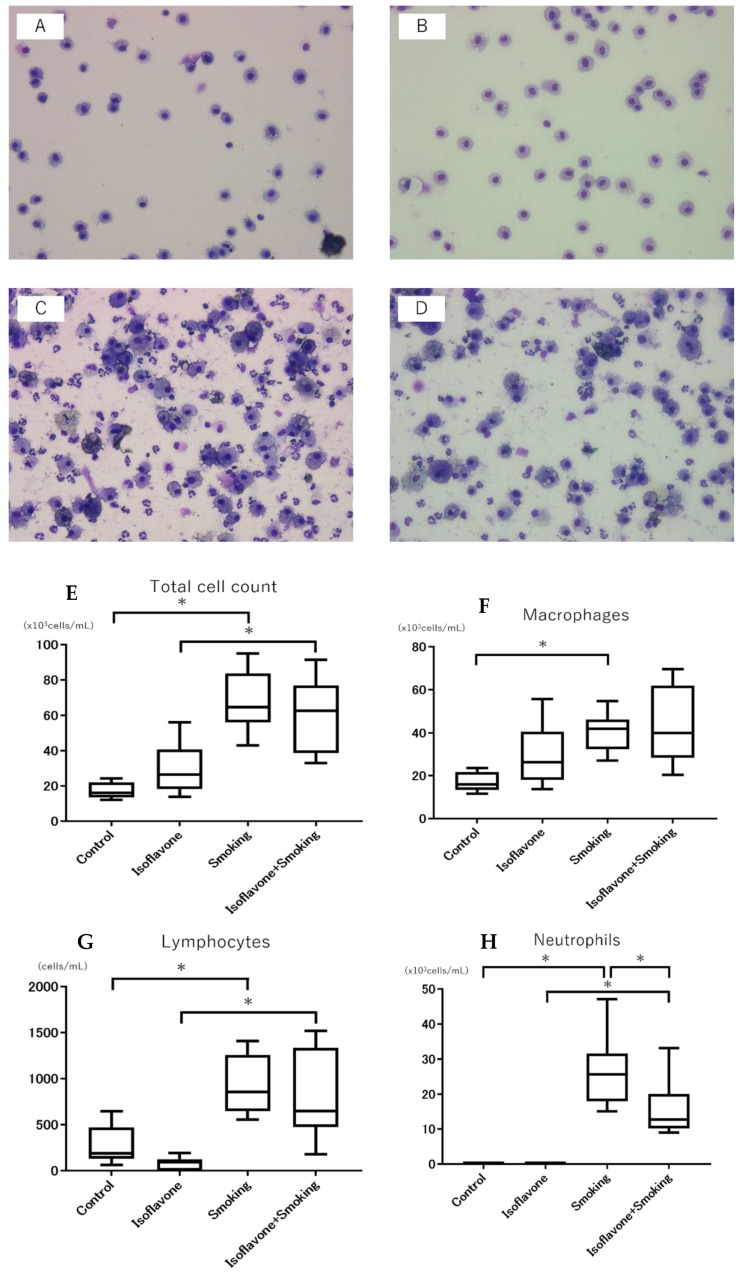
Representative micrographs of the bronchoalveolar lavage fluid (BALF) stained with the Diff-Quick. Images of (**A**) control group, (**B**) isoflavone group, (**C**) smoking group, and (**D**) isoflavone + smoking group are shown at 200× magnification. The cell counts of (**E**) total cells, (**F**) macrophages, (**G**) lymphocytes, and (**H**) neutrophils in the BALF are shown. Cigarette smoke significantly increased the number of total cells, macrophages, neutrophils, and lymphocytes. Isoflavone aglycones significantly decreased neutrophil cell counts. Data are shown with box and whisker plots. * *p* < 0.05.

**Figure 3 nutrients-11-02023-f003:**
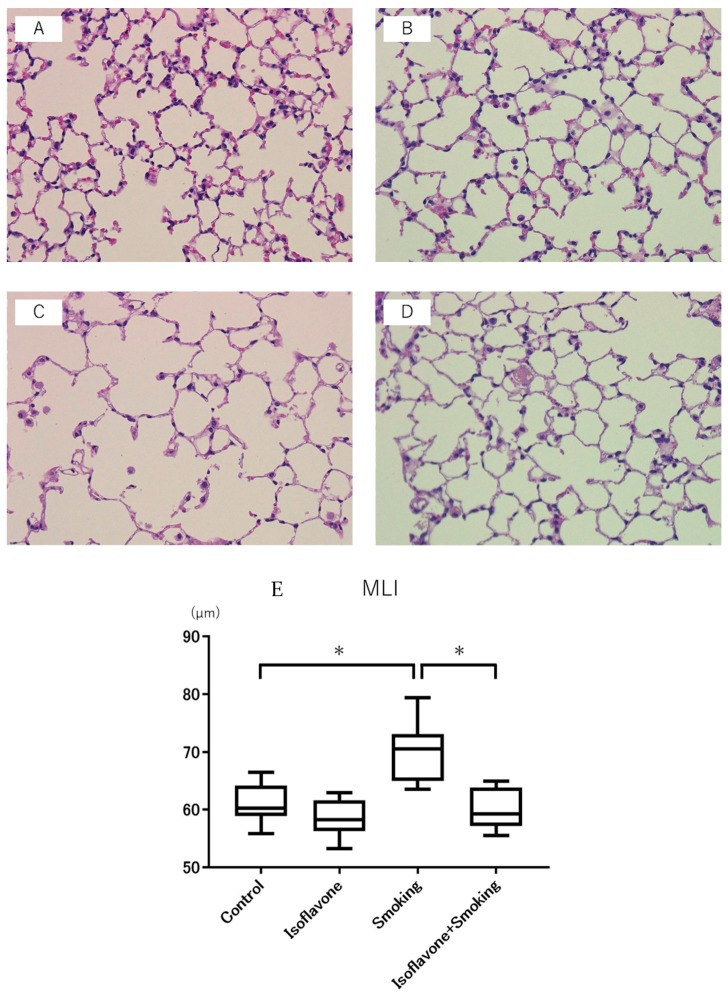
Representative micrographs of mice lungs stained with hematoxylin and eosin. Images of (**A**) control group, (**B**) isoflavone group, (**C**) smoking group, and (**D**) isoflavone + smoking group are shown at 200× magnification. (**E**) Mean linear intercepts (MLI) in each group. Isoflavone significantly decreased the MLI. Data are shown with box and whisker plots. * *p* < 0.05.

**Figure 4 nutrients-11-02023-f004:**
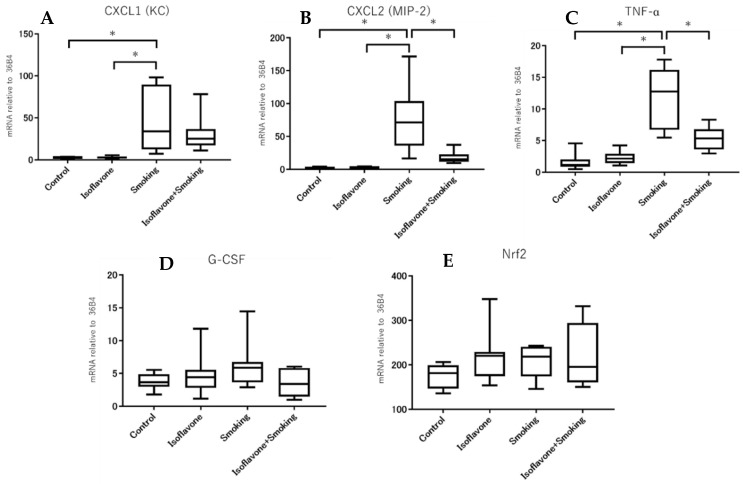
Relative mRNA level of each cytokine, chemokine and Nrf2. The mRNA level of (**A**) CXCL1 (KC), (**B**) CXCL2 (MIP-2), (**C**) TNF-α, (**D**) G-CSF, and (**E**) Nrf2 are shown. Cigarette smoke exposure increased gene-expression of TNF-α, CXCL1 (KC), and CXCL2 (MIP-2). Isoflavone significantly decreased the cigarette smoke-induced TNF-α and CXCL2 (MIP-2) gene expression. Data are shown with box and whisker plots. * *p* < 0.05.

**Figure 5 nutrients-11-02023-f005:**
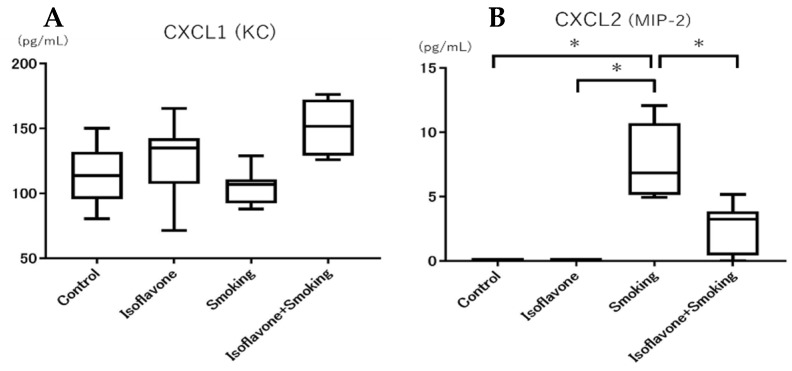
Protein level of CXCL1 (KC) and CXCL2 (MIP-2) in the BALF. (**A**) CXCL1 (KC) and (**B**) CXCL2 (MIP-2). Isoflavone significantly decreased the cigarette smoke-induced CXCL2 (MIP-2) level. Data are shown with box and whisker plots. * *p* < 0.05.

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
