# Peer review of "Isoflavone Aglycones Attenuate Cigarette Smoke-Induced Emphysema via Suppression of Neutrophilic Inflammation in a COPD Murine Model"

_nutrients, 2019, doi:10.3390/nu11092023_

Round 1
Reviewer 1 Report
This manuscript examines the effect of daidzein-rich soy isoflavone aglycones on inflammation and airspace enlargement in a murine model of COPD. I found the study results to be interesting, however I do have a number of queries I would like addressed:
· Why is group 2 (isoflavone) not statistically significantly different to group 3 (smoking) in Figures 4 and 5? Visually it looks like it should be. Was this study underpowered?
· Why were the numbers of mice/group not evenly spread across groups?
· Statistical analysis – were all the data normally distributed? Was normality tested?
· Line 56-58 – The studies you refer to look at isoflavones in terms of lung function and COPD risk, not neutrophilic inflammation. It is therefore not possible to state that isoflavones have anti-inflammatory effects and protect against neutrophilic inflammation.
· Did you measure plasma levels of daidzein and genistein in the mice before vs after supplementation? This would be worth reporting to confirm levels have changed. You could also look at correlations between changes in levels of these vs changes in inflammation etc.
· How does the level of supplementation given to the mice compare to a human diet? i.e. is it humanly possible to achieve these levels through the diet? How many soybeans would need to be consumed to reach levels these mice were receiving through supplementation?
· “Emphysema levels” (line 19, abstract) – what is this specifically? It sounds quite vague, I would re-word.
· Daidzein is miss-spelt in the abstract (line15).
· Why is genistein mentioned in the abstract but not the Introduction?
· Title – should probably read “…..neutrophilic inflammation in a COPD murine model”.
· Line 45 – would be better worded as “mediators that promote inflammation, which can contribute to the development of emphysema”.
· Line 52 – would be better worded as “Isoflavones are polyphenolyic compounds that exist in a number of foods including soybeans”, as soybeans aren’t the only source.
· The study described in line 54-55 needs to be referenced.
· Some of the tense throughout is incorrect, e.g. line 118 – “Body weight for each group is shown in Figure 1”. There are also a few typographical errors throughout that should be corrected (e.g. line 219 “underlying” rather than “underlining”).
· Line 196 – “…found primarily in soy products such as ….”
· Line 199-201 I feel is over-stated. Rather than providing “direct proof”, this study provides a “proposed mechanism”.
Author Response
Reviewer 1:
·Why is group 2 (isoflavone) not statistically significantly different to group 3 (smoking) in Figures 4 and 5? Visually it looks like it should be. Was this study underpowered?
Thank you for your suggestion. As reviewer pointed out, there is significant difference between group 2 and 3 in Figures 4 and 5. We have changed the figures.
·Why were the numbers of mice/group not evenly spread across groups?
Thank you for your suggestion. At the beginning, each group consisted of ten mice. However, we failed to collect some samples such as BALF, blood, and lung tissues by technical reasons. Therefore, there were missing data and the number of mice/group were not evenly spread.
· Statistical analysis – were all the data normally distributed? Was normality tested?
Thank you for your suggestion. All the data were re-analyzed by the Kolmogorov-Smirnov test. And the data were normally distributed. (pages 3, lines 115)
· Line 56-58 – The studies you refer to look at isoflavones in terms of lung function and COPD risk, not neutrophilic inflammation. It is therefore not possible to state that isoflavones have anti-inflammatory effects and protect against neutrophilic inflammation.
We thank the reviewer’s comment and apologize for making you confused. These reports which we referred were including the references no. 12-18. Therefore, we stated that isoflavones have anti-inflammatory effects and protect against neutrophilic inflammation. However, based on your suggestion, we have revised the word from “cigarette smoke-induced neutrophilic inflammation” to “cigarette smoke-induced inflammation.” (pages 2, lines 53-61)
· Did you measure plasma levels of daidzein and genistein in the mice before vs after supplementation? This would be worth reporting to confirm levels have changed. You could also look at correlations between changes in levels of these vs changes in inflammation etc.
· How does the level of supplementation given to the mice compare to a human diet? i.e. is it humanly possible to achieve these levels through the diet? How many soybeans would need to be consumed to reach levels these mice were receiving through supplementation?
Thank you for your suggestion. Unfortunately, we did not measure plasma levels of daidzein and genistein. Therefore, we could not evaluate the correlations between changes in levels of these vs changes in inflammation etc. However, regarding isoflavone dose, previous studies reported that 0.5% of soy isoflavone in the diet for mice is similar to the dose of isoflavone that human dose of 810mg/50kg body weight per day as below. And it is the amount that human consume in the dietary supplement. Thus, human can achieve these levels though the diet [1-3]. We have added words to the limitation. (pages 10, lines 298-301)
[1] Tabata, S.; Aizawa, M.; Kinoshita, M.; Ito, Y.; Kawamura, Y.; Takebe, M.; Pan, W.; Sakuma, K. The influence of isoflavone for denervation-induced muscle atrophy. In Eur J Nutr, Germany, 2019; Vol. 58, pp. 291-300.
[2] Bloedon, L.T.; Jeffcoat, A.R.; Lopaczynski, W.; Schell, M.J.; Black, T.M.; Dix, K.J.; Thomas, B.F.; Albright, C.; Busby, M.G.; Crowell, J.A., et al. Safety and pharmacokinetics of purified soy isoflavones: single-dose administration to postmenopausal women. Am J Clin Nutr 2002, 76, 1126-1137, doi:10.1093/ajcn/76.5.1126.
[3] Reagan-Shaw, S.; Nihal, M.; Ahmad, N. Dose translation from animal to human studies revisited. In FASEB J, United States, 2008; Vol. 22, pp. 659-661.
· “Emphysema levels” (line 19, abstract) – what is this specifically? It sounds quite vague, I would re-word.
Thank you for your suggestion. We have revised the word from “emphysema levels” to “airspace enlargement.” (pages 1, lines 20-21)
· Daidzein is miss-spelt in the abstract (line15).
Thank you for your suggestion. As the reviewer pointed out, we have corrected the word. (pages 1, lines 16)
· Why is genistein mentioned in the abstract but not the Introduction?
Thank you for your suggestion. As the reviewer pointed out, we have mentioned about genistein in the introduction. (pages 2, lines 54)
· Title – should probably read “…..neutrophilic inflammation in a COPD murine model”.
Thank you for your suggestion. As the reviewer pointed out, we have revised title from “in COPD murine model” to “in a COPD murine model.” (pages 1, lines 4)
· Line 45 – would be better worded as “mediators that promote inflammation, which can contribute to the development of emphysema”.
Thank you for your suggestion. As the reviewer pointed out, we have revised word from “mediators that promote inflammation leading to emphysema” to “mediators that promote inflammation, which contribute to the development of emphysema.” (pages 2, lines 46)
· Line 52 – would be better worded as “Isoflavones are polyphenolyic compounds that exist in a number of foods including soybeans”, as soybeans aren’t the only source.
Thank you for your suggestion. As the reviewer pointed out, we have revised word from”in soybeans” to “in a number of foods including soybeans.” (pages 2, lines 53)
· The study described in line 54-55 needs to be referenced.
Thank you for your suggestion. As the reviewer pointed out, we have added a reference. (pages 2, lines 57)
· Some of the tense throughout is incorrect, e.g. line 118 – “Body weight for each group is shown in Figure 1”. There are also a few typographical errors throughout that should be corrected (e.g. line 219 “underlying” rather than “underlining”).
Thank you for your suggestion. As the reviewer pointed out, we have corrected the tense and a typographical error. (pages 3, lines 120 and page 8, lines 236)
· Line 196 – “…found primarily in soy products such as ….”
Thank you for your suggestion. As the reviewer pointed out, we have added “primarily” in the sentence. (pages 8, lines 212)
· Line 199-201 I feel is over-stated. Rather than providing “direct proof”, this study provides a “proposed mechanism”.
Thank you for your suggestion. As the reviewer pointed out, we have revised word from “direct proof” to “proposed mechanism.” (pages 8, lines 216)

Reviewer 2 Report
This manuscript describes and investigates the protective effects of daidzein-rich soy isoflavone aglycones (DRIAs) in a cigarette smoke-induced experimental model of emphysema in mice. Dietary supplementation with 0.6% DRIAs attenuated cigarette smoke-induced pathogenesis by measuring mean linear intercept and inflammation. Although this work demonstrates an association between DRIAs and a cigarette smoke-induced experimental model of emphysema, there are no mechanistic data at the cellular or molecular levels and the manuscript presents as a survey of generic markers instead of a hypothesis-driven design.
Major Concerns
1. The study only used male mice and does not attempt to discuss any gender-specific responses in regard to both cigarette smoke and DRIA supplementation.
2. It is not clear why the data are presented as mean±SEM and not mean±SD. Based on the experimental design, SD should be used since it will describe variability amongst individual replicates while SEM represents an estimate of the population mean.
3. Why is it that the representative tissue images in Figure 3 do not show any signs of inflammation when Figure 2 depicts cigarette smoke-induced inflammation?
4. If KC and MIP-2 are both secreted by macrophages and there is no DRIA-dependent change in macrophage numbers, why would DRIAs selectively reduce cigarette smoke-induction of MIP-2?
5. Analyzing Nrf2 status by transcript levels is inappropriate. There is abundant literature on Nrf2 regulation and most studies analyze Nrf2 by subcellular localization and/or transcriptional activity. It would be much more appropriate to look at transcript levels of Nrf2-dependent genes, such as NQO1, GST, TXNRD1, and HO-1, in these tissue samples.
6. It is a very ambitious conclusion from the reported data that DRIA attenuated neutrophilic inflammation occurs via reductions of TNFα and MIP-2 without mechanistic and longitudinal studies defining this process. For example, it is not known which cells are primarily responsible for producing these cytokines and if they are direct or indirect target of DRIAs.
Minor Concerns
1. Unclear of timing of DRIA-supplemented diet. Was this started prior to cigarette smoke challenge or did they coincide?
2. What was the minimal number of cells from BALF quantified for differential counts?
3. Although significant differences in overall body weight were detected between control and cigarette smoke-treated cohorts, a longitudinal analysis should also depict the age(s) where body weight was significantly different.
4. Figures 4 and 5 appear out of order.
Author Response
Reviewer 2:
Major Concerns
1. The study only used male mice and does not attempt to discuss any gender-specific responses in regard to both cigarette smoke and DRIA supplementation.
We thank the reviewer’s comment. As the reviewer pointed out, the epidemiological study which we referred was reported about both genders. However, epidemiologically, male patients predominate the COPD population and this cigarette smoke-induced model for three months originally use the male mice. Thus, we used male mice in this study. Gender-specific response is also important issue. Based on your suggestion, we have added the sentence to the limitation (pages 10, lines 294-296), and we would consider the model using female mice in the future.
2. It is not clear why the data are presented as mean±SEM and not mean±SD. Based on the experimental design, SD should be used since it will describe variability amongst individual replicates while SEM represents an estimate of the population mean.
We thank the reviewer’s comment. As reviewer pointed out, we changed the data and figures to present as mean±SD. (pages 3, lines 115 and page 4, line 130)
3. Why is it that the representative tissue images in Figure 3 do not show any signs of inflammation when Figure 2 depicts cigarette smoke-induced inflammation?
Thank you for your suggestion. This cigarette-smoke exposed mice model can be modelled neutrophilia in BALF and emphysema. However, basically, chronic bronchitis with neutrophilic infiltration can not be modelled in this model[1]. Therefore, we could not show signs of inflammation in figure 3.
[1] Vlahos, R.; Bozinovski, S. Recent advances in pre-clinical mouse models of COPD. Clin Sci (Lond) 2014, 126, 253-265.
4. If KC and MIP-2 are both secreted by macrophages and there is no DRIA-dependent change in macrophage numbers, why would DRIAs selectively reduce cigarette smoke-induction of MIP-2?
We thank the reviewer’s comment, what you concerned is important. As we had discussed it in the discussion section (pages 9, lines 260-272), the difference between CXCL1(KC) and MIP-2 may be related the response to the LPS in cigarette smoke. Further in vivo and in vitro model may reveal the detailed mechanisms. Unfortunately, to the best of my knowledge, this is the potential reason that we can explain about the difference so far.
5. Analyzing Nrf2 status by transcript levels is inappropriate. There is abundant literature on Nrf2 regulation and most studies analyze Nrf2 by subcellular localization and/or transcriptional activity. It would be much more appropriate to look at transcript levels of Nrf2-dependent genes, such as NQO1, GST, TXNRD1, and HO-1, in these tissue samples.
Thank you for your suggestion. As reviewer pointed out, the assessment of Nrf2 pathway activation such as Nrf2 subcellular localization or Nrf2 targeted gene expression is more important for antioxidant capacity. Nrf2 is known to be upregulated by various stimulation and total Nrf2 might be a surrogate marker of nuclear translocated Nrf2 level or Nrf2 pathway activation. Regrettably, we assessed total Nrf2 level, not nuclear translocated Nrf2 level, in this study. Thus, we have softened the expression in the discussion (pages 10, lines 289-290).
6. It is a very ambitious conclusion from the reported data that DRIA attenuated neutrophilic inflammation occurs via reductions of TNFα and MIP-2 without mechanistic and longitudinal studies defining this process. For example, it is not known which cells are primarily responsible for producing these cytokines and if they are direct or indirect target of DRIAs.
We thank reviewer’s comment. As reviewer pointed out, we agree that the mechanistic and longitudinal studies are important to reveal direct relationship between DRIAs and the attenuation of neutrophilic inflammation. Unfortunately, we could not clarify the mechanism in a vitro model, however, out data supports the previous epidemiological data that isoflavones have protective effects to prevent COPD, and we could expand the possibility to extend a way for COPD treatment and prevention in this study. Thus, we have softened the conclusion (pages 10, lines 303-304).
Minor Concerns
1. Unclear of timing of DRIA-supplemented diet. Was this started prior to cigarette smoke challenge or did they coincide?
We thank the reviewer’s comment. Mice started to receive each diet when mice were randomized into groups. Therefore, the timing was prior to cigarette smoke challenge. Based on your suggestion, we have added the sentence to the material and methods. (pages 2, lines 75-76)
2. What was the minimal number of cells from BALF quantified for differential counts?
Thank you for your suggestion. Differential counts in BALF of human are generally performed by counting at least 400 cells [2]. Therefore, we counted more than 400 cells. We have added the sentence (pages 2, lines 87).
[2] Meyer, K.C.; Raghu, G.; Baughman, R.P.; Brown, K.K.; Costabel, U.; du Bois, R.M.; Drent, M.; Haslam, P.L.; Kim, D.S.; Nagai, S., et al. An official American Thoracic Society clinical practice guideline: the clinical utility of bronchoalveolar lavage cellular analysis in interstitial lung disease. In Am J Respir Crit Care Med, United States, 2012; Vol. 185, pp. 1004-1014.
3. Although significant differences in overall body weight were detected between control and cigarette smoke-treated cohorts, a longitudinal analysis should also depict the age(s) where body weight was significantly different.
We thank the reviewer’s comment. There is no significant difference in the body weight at pretreatment and 1week. However, there are significant differences in the body weight between control group and isoflavone group, and smoking group and isoflavone group after 2 weeks. We have added the sentence in the legend for figure1. (pages 4, lines 130-132)
4. Figures 4 and 5 appear out of order.
We thank the reviewer’s comment. We have revised figures 4 and 5. (pages 7 and 8)

Round 2
Reviewer 2 Report
I appreciate the author's responses and changes reflected in the manuscript.
